# Prominin-1 Knockdown Causes RPE Degeneration in a Mouse Model [note 1]

**DOI:** 10.3390/cells13211761

**Published:** 2024-10-24

**Authors:** Sujoy Bhattacharya, Tzushan Sharon Yang, Bretton P. Nabit, Evan S. Krystofiak, Tonia S. Rex, Edward Chaum

**Affiliations:** 1Department of Ophthalmology and Visual Sciences, Vanderbilt University Medical Center, Nashville, TN 37232, USAedward.chaum@vumc.org (E.C.); 2Pathology, Microbiology, and Immunology, Vanderbilt University Medical Center, Nashville, TN 37232, USA; 3Cell and Developmental Biology, Vanderbilt University, Nashville, TN 37232, USA

**Keywords:** atrophic age-related macular degeneration, geographic atrophy, adeno-associated virus (AAV2/1), lysosomal pathways, mitochondria, microglia

## Abstract

There are currently no effective treatments for retinal pigment epithelial (RPE) cell loss in atrophic AMD (aAMD). However, our research on *Prominin-1* (*Prom1*), a known structural protein in photoreceptors (PRs), has revealed its distinct role in RPE and offers promising insights. While pathogenic *Prom1* mutations have been linked to macular diseases with RPE atrophy, the broader physiological impact of dysfunctional *Prom1* in RPE loss is unclear. We have shown that *Prom1* plays a *crucial* role in regulating autophagy and cellular homeostasis in *human* and *mouse* RPE (mRPE) cells in vitro. Nevertheless, a comprehensive understanding of its in vivo expression and function in mRPE remains to be elucidated. To characterize *Prom1* expression in RPE in situ, we used RNAscope assays and immunogold electron microscopy (EM). Our use of chromogenic and fluorescent RNAscope assays in albino and C57BL/6J *mouse* retinal sections has revealed *Prom1* mRNA expression in perinuclear regions in mRPE in situ. Immunogold EM imaging showed *Prom1* expression in RPE cytoplasm and mitochondria. To confirm *Prom1* expression in RPE, we interrogated *human* RPE single-cell RNA-sequencing datasets using an online resource, Spectacle. Our analysis showed *Prom1* expression in *human* RPE. To investigate *Prom1*’s function in RPE homeostasis, we performed RPE-specific *Prom1* knockdown (KD) using subretinal injections of AAV2/1.CMV.saCas9.U6.*Prom1*gRNA in male and female *mice*. Our data show that RPE-specific *Prom1*-KD in vivo resulted in abnormal RPE morphology, subretinal fluid accumulation, and secondary PR loss. These changes were associated with patchy RPE cell death and reduced a-wave amplitude, indicating retinal degeneration. Our findings underscore the central role of *Prom1* in cell-autonomous mRPE homeostasis. The implications of *Prom1*-KD causing aAMD-like RPE defects and retinal degeneration in a *mouse* model are significant and could lead to novel treatments for aAMD.

## 1. Introduction

The Prominin-1 (*Prom1*) gene encodes a transmembrane glycoprotein [1,2], which is widely recognized as an antigenic marker for stem cells and cancer stem cells [3,4]. *Prom1* is expressed in differentiated epithelial and non-epithelial cells [5], glial cells [5], and the adult retina [6], suggesting that *Prom1* plays a general role beyond stemness and differentiation status and is not limited to specific cell types. In the retina, *Prom1* is present in the photoreceptor outer segments [7] and plays an essential role in photoreceptor disk morphogenesis [8]. The loss-of-function *Prom1* mutations result in inherited retinal dystrophies, including autosomal dominant and autosomal recessive retinitis pigmentosa [7,9], cone–rod dystrophies [10,11,12], and macular dystrophies [8,13]. *Prom1*-associated macular dystrophy, also known as Stargardt disease 4 (STGD4), has clinical and pathophysiological features similar to ABCA4-related Stargardt disease 1 (STGD1) and the atrophic (dry) form of age-related macular degeneration (AMD), where abnormal cytotoxic lipofuscin bisretinoid accumulation triggers degeneration of macular rods, cones, and retinal pigment epithelial (RPE) cells [14,15,16]. This indicates that *Prom1* dysfunction causes photoreceptor and RPE degeneration, primarily in the macula, but that disease progression may or may not involve increased bisretinoid lipofuscin. Understanding how *Prom1* regulates RPE health and homeostasis is crucial for developing effective therapies.

We have previously shown that *Prom1* is expressed in *human* and *mouse* RPE cell cultures in vitro. Somewhat surprisingly, *Prom1* is predominantly a cytoplasmic protein in *human* RPE cells, and *Prom1* knockout (KO) in *human* RPE cells activates mTORC1/2 activities and impairs the trafficking of autophagosomes to lysosomes [17]. Our studies also showed that cytosolic *Prom1* interacts with p62 and HDAC6 in the developing autophagosome in *human* RPE cells, confirming its role in autophagy regulation [17]. We have recently shown that *Prom1*-KO activates mTORC1, reduces TFEB activity, and induces epithelial–mesenchymal transition (EMT) in *mouse* RPE (mRPE) cells, demonstrating that *Prom1*-mTORC1-TFEB signaling is a central driver of cell-autonomous mRPE homeostasis and suggesting a possible role in the development of geographic atrophy (GA) [18].

*Prom1*-related retinopathies are associated with various pathogenic *Prom1* variants and heterogeneous phenotypical characteristics. The main phenotypic distinction lies between recessive and dominant forms of the disease [19]. While the recessive disease is associated with early-onset retinal degeneration, the dominant disease is associated with late-onset dystrophy, predominantly involving the macula, demonstrating that *Prom1* mutations and inheritance patterns differentially impact multiple cell types in the outer retina. It is unclear whether *Prom1* dysfunction primarily affects the photoreceptor cells with secondary involvement of the RPE or whether the RPE is a primary origin of the disease. Increasing evidence suggests that loss-of-function *Prom1* mutations, including c.400C>T, p.R373C, and c.869delG mutants, cause RPE granular mottling, thinning of the outer retina, and parafoveal RPE atrophy in the macula [20,21,22]. In some younger patients with a mean age of 42 years, SD-OCT showed thinning of the RPE/Bruch’s membrane, indicative of RPE cell loss and early progression to GA [23]. Consistent with these observations, ophthalmic findings in younger patients with *Prom1* R373C mutation exhibit distinct macular phenotypes, including central GA, multifocal GA, and bull’s eye maculopathy [24]. In subsequent follow-up studies of these young patients, the GA area was significantly enlarged in a time-dependent manner. Profound degeneration of the outer retinal layer, accompanied by extensive loss of RPE cells, was consistent with the longitudinal progression of *Prom1*-associated retinal degeneration. While these findings suggest that *Prom1* dysfunction primarily impacts the RPE, additional investigations are necessary to understand how *Prom1* function differentially regulates photoreceptor versus RPE homeostasis.

To address this knowledge gap, this study focused on understanding *Prom1*’s expression and its significance in RPE biology. To validate *Prom1* expression in the pigmented RPE in situ, we used chromogenic and fluorescent RNAscope assays and immunogold electron microscopy. To demonstrate that loss of *Prom1* in the *mouse* leads to RPE degeneration in vivo and recapitulates clinical features of atrophic AMD, we used AAV2/1-mediated *Prom1* gene knockdown (KD). Our studies demonstrate that *Prom1* plays distinct roles in the photoreceptors vs. RPE and begin to demonstrate how *Prom1* preserves the physiologic functions of the RPE.

## 2. Materials and Methods

### 2.1. Materials

The backbone replication-deficient all-in-one purified experimental viral particles AAV2/1.CMV.saCas9.U6.*Prom1*gRNA (titer of 6.76 × 10^22^ Genome Copies/mL) and control viral particles AAV2/1.CMV.saCas9.U6.scrambledgRNA (titer of 7.96 × 10^12^ Genome Copies/mL) gRNA were commercially obtained from GeneCopoeia (Rockville, MD, USA). The AAV particles were generated following a standardized protocol using highly purified plasmids and Endofectin-AAV reagents. The mouse *Prom1*-transcript variant 4 mRNA-RNAscope probe (catalog # 412221) and RNAscope assay kit were purchased from ACDBio (Newark, CA, USA). The cleaved caspase-3 (Asp175) (D3E9) rabbit mAb- Alexa Fluor 647 conjugate (catalog# 9602), goat serum (catalog # 5425), and Prolong Gold AntiFade Reagent with DAPI (catalog #8961), anti-p62 antibody (catalog# 23214), and anti-LC3-I/II antibody (catalog# 12741) were obtained from Cell Signaling Technology (Danvers, MA, USA).

### 2.2. Mice and Colony Management

C57BL/6J *mice* were obtained from the Jackson Laboratory (Bar Harbor, ME, USA) (JAX, stock #000664), and the albino sentinel mice were from Charles River (Wilmington, MA, USA) (CD-1 strain code 022). *Mice* were housed, maintained on a 12 h lightdark cycle, and provided food and water ad libitum. The Institutional Animal Care and Use Committee of Vanderbilt University Medical Center (VUMC) approved all experiments. All animal procedures followed the guidelines of the Association for Research in Vision and Ophthalmology Statement on the Use of Animals in Ophthalmic and Vision Research. Both male and female *mice* (4–6 weeks old) were used for this project.

### 2.3. Subretinal Injections

Both male and female C57BL/6J *mice* (4–6 weeks of age) were anesthetized by intraperitoneal injection with a mixture of 12.5 mg/Kg xylazine and 62.5 mg/kg ketamine. Topical ocular anesthesia was performed using 0.5% proparacaine, and the pupils were dilated with 0.5% tropicamide. Following dilation, the animals were placed under a dissecting microscope (Nikon, Tokyo, Japan), and the fundus was visualized with a drop of 2.5% methylcellulose. The sclera was punctured posterior to the limbus with a 33-gauge hypodermic Hamilton needle, avoiding trauma to the iris and the lens. The needle was placed at the inferior site of the ora serrata and advanced transsclerally into the subretinal space, as described earlier [25]. The contents of the syringe, 1 mL of the viral vector solution (AAV2/1-saCas9-*Prom1*-gRNA or AAV2/1-saCas9-scrambled-gRNA), were slowly released into the subretinal space.

### 2.4. Electroretinogram (ERG) in Mice

*Mice* were dark-adapted overnight, dilated with 1% tropicamide, and anesthetized with 20/8/0.8 mg/kg ketamine/xylazine/urethane according to previously published methods [26]. To prevent hypothermia in anesthetized *mice*, they were placed on the heated surface of the ERG system. Corneal electrodes with integrated stimulators (Celeris System, Diagnosys LLC), (Lowell, MA, USA) were placed on eyes lubricated with GenTeal drops. The subdermal platinum needle electrodes were placed in the snout and back of the head at the location of the visual cortex. A ground electrode was placed in the back of the mouse. For ERGs, *mice* were exposed to 50 flashes of 1 Hz, 0.05 cd s/m^2^ white light with a pulse frequency of 1, as published earlier [26]. Each experimental group had 13–15 eyes.

### 2.5. Chromogenic and Fluorescent Prom1 RNAscope Assays in Mouse Retina Sections

The freshly isolated eyeballs from the euthanized C57BL6J and sentinel *mice* were fixed in 10% neutral-buffered formalin (NBF) overnight, embedded in paraffin, cut into 3.5–4 mm sections, and placed on glass slides. In some cases, the ocular posterior cup was dissected from the *mouse* eye for in situ localization of *Prom1* in *mouse* RPE. The slides were then placed on the Leica Bond-RX IHC stainer, and all steps besides coverslipping were performed on the Bond IHC stainer. Slides were baked and deparaffinized, and heat-induced antigen retrieval was performed using the Lecia Epitope Retrieval 2 solution at 95 °C and the ACD 2.5 LSx protease enzyme using the ACD RNAscope^®^ protocol. Slides were hybridized with the *Prom1* probe (Cat#412228, 23214, ACD-Bio-Techne (Newark, CA, USA) for 2 h. Both negative (Dapb) and positive (PPIB) control slides were used alongside the target probe. The ACD RNAscope^®^ 2.5 LSx Reagent Kit- chromogenic RED detection system was used to visualize RPE and PRs of the sentinel albino *mouse* retina sections. The slides were counterstained with hematoxylin, a Leica Bond detection system component. Slides were dehydrated, coverslipped, and used for a 40X brightfield Leica SCN400 slide scanner at Vanderbilt’s Digital Histology Shared Resource (DHSR). The images were analyzed using the Aperio ImageScope (v.12.4.6.5003) available through the Digital Slide Archive (DSA) at Vanderbilt.

To visualize *Prom1* expression in pigmented RPE in the C57/BL6J *mice*, we used the ACD RNAscope^®^ Multiplex Fluorescent Reagents Kit system containing OPAL570 and Cy5 fluorescent markers. DAPI was used as a counterstain to visualize nuclei, an ACD/Leica RNAscope kit component. Slides were coverslipped, and images were taken using the Nikon widefield microscope (Tokyo, Japan) (100X objective) and the Zeiss LSM 880 confocal microscope (Oberkochen, Germany) (60X objective) at the Cell Imaging Shared Resource at Vanderbilt. Images were also captured using a 40× Leica fluorescent whole slide imager at Vanderbilt’s DHSR.

### 2.6. H&E Staining and Histology

*Mice* subretinally injected with scrambled or *Prom1*-gRNA were housed and fed ad libitum at the Vanderbilt University Department of Animal Care. After 10–11 weeks post-injection, *mice* were euthanized by CO_2_ asphyxiation followed by thoracotomy, and eyes were collected and fixed in 4% PFA. Eyes were subsequently embedded in paraffin, cut into 3.5–4 mm sections, and stained with hematoxylin–eosin (H&E). The stained sections were analyzed using the Translational Pathology Shared Resource (TPSR), and images were captured using a high-throughput Leica SCN400 40× Brightfield slide scanner automated digital image system from Leica Microsystems at Vanderbilt’s Digital Histology Shared Resource (DHSR). All whole slides were imaged at 40× magnification to a 0.25 mm/pixel resolution. The slides were viewed and annotated, and images were analyzed using the Aperio ImageScope (v.12.4.6.5003) available through Vanderbilt’s DSA.

### 2.7. Immunohistochemistry

The formalin-fixed-paraffin-embedded *mouse* retinal sections were placed on slides, and all steps involving immunohistochemistry (IHC) were performed on the Bond IHC stainer at Vanderbilt’s TPSR. Slides were deparaffinized, and heat-induced antigen retrieval was performed using the Epitope Retrieval 1 solution for 20 min. Slides were then incubated with anti-LC3A/B (Cat#23214, Cell Signaling, Danvers, MA, USA) and anti-p62 (Cat#23214, Cell Signaling, Danvers, MA, USA) for 1 h at dilutions of 1:2500 and 1:500, respectively. The Bond Polymer Refine Detection system was used for chromogenic visualization. The slides were dehydrated, coverslipped with hematoxylin as a counterstain, and scanned using the Leica 40X SCN400 Brightfield scanner at DHSR. Images were analyzed using the Aperio ImageScope available through DSA at VUMC.

To detect active caspase-3 levels in *mouse* retinal sections, antigen retrieval was performed, and the slides were blocked with goat serum. Subsequently, the slides were incubated overnight with an anti-cleaved-caspase3-Alexa Fluor 647 conjugate antibody at 4 °C. The slides were washed three times with 1X PBS and coverslipped with an antifade mounting medium containing DAPI. The fluorescent immunostained tissue slides were imaged on an Aperio Versa 200 automated slide scanner (Leica Biosystems, Danvers, MA, USA) at 40X magnification to a 0.162 mm/pixel resolution. The images were analyzed using the Aperio ImageScope at Vanderbilt’s DHSR.

### 2.8. Transmission Electron Microscopy (TEM)

TEM was performed using methods described earlier with modifications [27]. Briefly, eyecups were fixed in 4% PFA and 0.5% glutaraldehyde. After fixation, the samples were cryoprotected by gradual equilibrium with 30% glycerol followed by plunge freezing in liquid ethane. Samples were freeze-substituted in 1.5% uranyl acetate in methanol for 48 h at −80 °C, followed by infiltration with HM20 at −30 °C. The HM20 was polymerized with UV light for 48 h at −30 °C. Samples were sectioned at 100 nm nominal thickness of a UC7 ultramicrotome and collected onto 300 mesh Ni grids. For immunogold labeling, the grids were fixed, and antigens were retrieved using 0.1% sodium borohydride with 50 mM glycine and then blocked in 10% goat serum. Samples were incubated with rabbit polyclonal primary antibodies against *Prom1* from Origene (Rockville, MD, USA) (catalog number TA354470) and Abcam (Waltham, MA, USA) (catalog number ab19898) at 1:50 dilution for 2 h, followed by the secondary antibody at 1:20 for 1 h. The grids were poststained with 2% uranyl acetate. TEM imaging was performed on a JEOL 2100+ equipped with an AMT nanosprint15 MKII CMOS camera using AMT acquisition software (version number 702.97).

### 2.9. Analysis of Single-Cell RNA-Sequencing RPE and Retina Datasets

All published single-cell RNA-sequencing and ATAC *mouse* and *human* RPE/retina datasets were analyzed using Spectacle, an interactive online resource for single-cell RNA data analysis (https://singlecell-eye.org/app/spectacle/ (accessed on 15 May 2024)). Spectacle uses retina, RPE, and choroid datasets from *human* and *mouse* samples to identify which cell types express a gene of interest and characterize gene expression changes across regions or disease states [28].

### 2.10. Statistical Analysis

All data were analyzed using the GraphPad Prism 9 program (GraphPad Software Inc., San Diego, CA, USA). Data are expressed as mean ± S.E. An unpaired 2-tailed Student’s *t*-test and Bonferroni post hoc testing were used to assess statistical significance. Unless otherwise stated, * *p* < 0.05, ** *p* < 0.01, and **** *p* < 0.0001 values were considered significant.

## 3. Results

### 3.1. Expression of Prom1 in Mouse RPE In Situ

We have shown that *Prom1* is expressed in *human* and *mouse* RPE cells in vitro, but its expression in *mouse* RPE in situ has not been fully characterized beyond its immunoreactivity in the RPE microvilli [7,17,18]. Consistent with these observations, *Prom1* expression was detected in hTERT-RPE-1 cells and found to interact physically with rhodopsin, implying a role of *Prom1* in regulating RPE homeostasis [29]. To examine *Prom1* expression in *mouse* RPE in situ, we used a chromogenic *Prom1* RNAscope assay in albino *mouse* eyecups after removing the cornea, lens, and neuroretina. The RNAscope (RNA in situ hybridization) assay is used to visualize gene expression within tissues and detect specific RNA (mRNA) molecules and their spatial distribution [30]. Our results show *Prom1* mRNA expression in the *mouse* RPE in situ, as evidenced by the red chromogenic puncta in RPE (arrowheads) (Figure 1A). The mouse eyecup with the area used for imaging (red box) is shown in Figure 1B. However, *Prom1* labeling was not observed in the negative control, confirming the specificity of *Prom1* RNAscope (Figure 1C). To confirm *Prom1* mRNA expression, we repeated chromogenic RNAscope in another *mouse* eyecup. We found *Prom1* expression in RPE in situ (black arrowheads) (Figure 1D). The *mouse* eyecup used for this image is shown in Figure 1E (red box). To confirm its expression in non-pigmented *mouse* RPE, we used a fluorescent RNAscope assay (using Cy5 fluorophore) and found the localized and discrete puncta of *Prom1* mRNA in situ (white arrowheads) in the mouse eyecup (Figure 1F). Widefield high-resolution confocal microscopy showed *Prom1* magenta puncta in RPE (Figure 1G). The chromogenic RNAscope assay shows a strong *Prom1* signal (red signal) in the photoreceptor inner segments (ISs) (black arrowheads), outer nuclear layer (ONL), and, interestingly, a small cohort of cells in the inner retina (white arrowheads) (Figure 2A), but *Prom1* labeling was not observed in the negative control (Figure 2B). However, it is challenging to see *Prom1* in RPE using the chromogenic assay due to the pigmentation of RPE (Figure 2A). We used bleached retina sections to visualize *Prom1* in RPE but bleaching interfered with the chromogenic RNAscope signal. To address this, we used a fluorescent RNAscope with a Cys5 fluorophore and widefield microscopy to detect the *Prom1* signal in the pigmented RPE. Consistent with data on retinal sections from albino *mice*, we observed fluorescent *Prom1* magenta puncta in pigmented RPE from C57BL6J *mice*, confirming its expression (white arrows) (Figure 2C). Furthermore, a strong *Prom1* signal was observed in photoreceptor ISs and ONL, as well as a small group of cells in the inner retina (white arrowheads), but not in the negative control (Figure 2C,D). This confirms that the *Prom1* gene is highly expressed in the photoreceptors but is also detectable in RPE.

To visualize *Prom1* gene expression in single RPE cells in the C57BL6J retinal sections, we used 100X objective high-magnification widefield confocal microscopy. Our results show that the *Prom1* mRNA magenta puncta is detectable in single RPE cells in the *mouse* retinal sections (white arrowheads) (Figure 2E). We observed the *Prom1* signal near the DAPI-stained nuclei of RPE cells, predominantly localized in the basolateral region (Figure 2E). To further confirm *Prom1*’s expression in RPE, we used a different fluorophore, OPAL-570, which detects low-expressing mRNA, optimizes signal amplification, and reduces background noise [31]. The punctate *Prom1* mRNA expression (red) was detected in the pigmented RPE, predominantly in the perinuclear region of RPE cells (Figure 2F). Consistent with data presented in Figure 2E, the *Prom1* signal was mainly confined to the basolateral region of RPE cells and restricted from the apical processes (Figure 2F). These observations are consistent with our previous findings, showing that *Prom1* is a cytoplasmic protein that does not localize to the apical membranes in cultured *human* RPE in vitro [17].

To confirm *Prom1* expression in *mouse* RPE in situ, we used immunogold electron microscopy in C57BL/6J eyecups after removing the cornea, lens, and retina. Our data show sparse *Prom1* labeling in RPE cells using a rabbit polyclonal antibody from Origene. Analysis of *Prom1* labeling in RPE shows its presence in mitochondria (white arrows, Figure 3A,B). Additional immunogold TEM with an antibody from Aviva showed *Prom1*’s cytoplasmic localization in RPE (Figure 3C, white arrow, top left) and RPE mitochondria (Figure 3C, white arrow in middle left) and near mitochondria (Figure 3C, white arrow, bottom right). *Prom1*’s presence in mitochondria suggests it could influence mitochondrial dynamics, bioenergetics, and overall RPE health.

Since *Prom1* regulates autophagy in mRPE cells [18], we also examined the in situ localization of two key autophagy mediators, p62 and LC3I/II proteins, in *mouse* retinal sections. Chromogenic immunohistochemical (IHC) analysis of *mouse* retinal sections using p62 and LC3-I/II antibodies shows positive immunolabeling of p62 (Figure 4A) and LC3-I/II (Figure 4C) in RPE. However, no labeling was detected in their respective negative controls (Figure 4B,D). We found p62 immunoreactivity in the outer plexiform layer (OPL) and POS in the outer retina. Several other cell types in the inner retina showed p62 immunoreactivity, including the ganglion cell layer (GCL). We observed strong LC3-I/II immunoreactivity in the outer retina’s POS, ISs, ONL, OPL, and GCL in the inner retina. These results demonstrate that the critical autophagy regulators are detectable in the mRPE in situ.

### 3.2. AAV2/1-Mediated Prom1 Knockdown (KD) In Vivo Using CRISPR/Cas9-gRNA

Our data show that *mouse* eyes injected with AAV2/1-scambled gRNA did not alter *Prom1* expression in the RPE (Figure 5A–F, gray arrows, left panel). However, eyes injected with AAV2/1.CRISPR.*Prom1*.gRNA did not cause *Prom1* gene deletion throughout the entire RPE layer (Figure 5G–M, gray arrows, right panel), probably because the AAV transduction is limited to the injection site. We used low- and high-magnification images (with different scale bars) to focus on areas of RPE degeneration after subretinal injections of AAV2/1.CRISPR.*Prom1*.gRNA in the *mouse* eye. While low-magnification images provide a more comprehensive view of the entire retinal section (Figure 5A,B,G–I), allowing for the identification of general patterns and overall *Prom1* expression levels in control vs. *Prom1*.gRNA-injected eyes, magnified areas of the retinal sections enable precise localization and characterization of *Prom1* gene expression in *mouse* RPE in situ. We observed intact *Prom1* mRNA expression in retinal sections from *mouse* eyes injected with control AAV2/1.CRISPR.scr.gRNA (Figure 5C–F, gray arrows). Using images of different magnifications, we observed a *Prom1* (*Prom1*-KD) loss in localized areas through the RPE layer (Figure 5J–M, right panel, white arrowheads). Some areas showed intact *Prom1* expression, indicating that subretinal AAV2/1 cannot transduce the entire RPE layer (Figure 5J–M, gray arrows). Importantly, injection of AAV2/1-*Prom1*-gRNA did not cause downregulation of *Prom1* expression in the *mouse* PRs, demonstrating that AAV2/1 specifically targets *Prom1* in the non-dividing RPE without transducing other retinal cell types. There was no evidence of inflammatory response or subretinal neovascularization associated with subretinal delivery of AAV2/1-mediated gRNA. Some sections show artifactual detachment of the overlying retina with few areas of photoreceptor outer segments attached to the RPE.

AAV2 has been used effectively through subretinal injection to transduce the RPE in several gene therapy trials and animal models [32,33]. The subretinal injection of AAV2/1 into C57BL/6J *mice* can specifically target RPE genes without affecting other cell types in the murine retina [34,35]. To investigate if RPE-specific targeting of *Prom1* in a *mouse* model leads to RPE degeneration and secondary photoreceptor loss, we performed subretinal injections of control AAV2/1 carrying the scrambled gRNA and the experimental AAV2/1 carrying the *Prom1*-gRNA. The control vector did not alter the normal fundus (Figure 6A), whereas eyes injected with the *Prom1*-gRNA displayed cloudy, patchy, and diffuse yellowish lesions (Figure 6B,C) consistent with RPE cell pathology and/or dysfunction. Non-invasive in vivo scotopic full-field ERGs revealed waveform changes in *mouse* eyes injected with AAV2/1.CRISPR.*Prom1*.gRNA vs. control scrambled gRNA (Figure 6D). We observed a reduction in both the a- and b-wave amplitudes but focused on the a-wave. If the a-wave is compromised, the ON-bipolar cells will not receive the right input from the PRs’ initial signal to produce the b-wave. Therefore, the initial loss of the a-wave disrupts the signal transmission pathway, which leads to a subsequent reduction in the b-wave. Analysis of ERGs showed a significant decrease in a-wave (derived from the PR layer) amplitude in *mice* 11 weeks after subretinal injection with *Prom1*-gRNA (Figure 6E) as compared with those of *mice* injected with control gRNA, revealing functional abnormalities in PRs with RPE-specific *Prom1*-KD. These observations suggest that localized RPE dysfunction can lead to PR loss, impacting ERG readings in *mice*.

To determine the corresponding anatomical changes that occur with RPE pathology and loss of PR function, we evaluated the morphology of the retina in *mice* injected with either control or *Prom1*-gRNA. Light microscopy-based H&E sections show significant differences in AAV2/1*Prom1*-gRNA-injected eyes compared to control AAV2/1.scr.gRNA at 11 weeks post-injection (Figure 7). We observed RPE vacuoles and degeneration with fluid accumulation in the subretinal space causing PRs to be detached from the RPE in AAV2/1.*Prom1*-gRNA-injected (male and female) *mice* (Figure 7B,D), but not in age-matched *mice* injected with control AAV2/1.scr.gRNA (Figure 7A,C). These effects were not observed throughout the entire retina. Instead, they appeared as patches of RPE degeneration/loss in well-demarcated areas with PR degeneration.

### 3.3. RPE-Specific Prom1-KD in a Mouse Model Causes RPE Cell Death

Activation (i.e., cleavage) of caspase-3 has been implicated in RPE cell degeneration in aAMD [36]. Cell death mechanisms, including apoptosis and pyroptosis, are activated in rodent RPE cells in vivo in response to amyloid beta, a drusen component [37]. Elevated proteolytic cleavage of caspase-3, a marker for apoptosis, was observed in the degenerating *rodent* RPE [37]. We tested whether patchy *Prom1*-KD in response to AAV2/1-*Prom1* gRNA subretinal injection after 11 weeks is sufficient to trigger caspase-3 activation, leading to RPE death. Since RPE-specific *Prom1*-KD causes phenotypic changes in the RPE, resembling RPE degeneration, we tested if this is associated with RPE apoptosis. We used an antibody that detects active cleaved caspase-3 and found that subretinal injection of the control AAV1/2.CRISPR.scr.gRNA did not induce caspase-3 activation (Figure 8A,E, left panel). However, eyes injected with AAV2/1 carrying *Prom1*-gRNA showed active caspase-3 positive RPE cells (Figure 8B,F, right panel, micrographs showing white arrows), but the PRs were largely unaffected. A few images in the sections from *Prom1*-KD *mouse* retinas, where the active caspase-3 antibody shows a diffuse background in the POSs, indicate that degenerating RPE can concomitantly damage PRs, causing PR apoptosis. It is unclear whether the diffuse caspase-3 signal in the POSs is due to PR death or non-specific staining, and further investigation is warranted. To demonstrate if *Prom1*-KD in these regions causes RPE cell death, we used serial sections from these areas and labeled them for *Prom1* using RNAscope. We found that the control AAV2/1.CRISPR.scr.gRNA did not affect *Prom1* gene expression (Figure 8C,G, left panel, white arrowheads). However, AAV2/1.CRISPR.*Prom1*.gRNA injection in *mouse* eyes showed loss of the *Prom1* gene in the areas where RPE cells were labeled positive for active cleaved caspase-3 (Figure 8D,H, right panel). These results demonstrate that RPE-specific *Prom1*-KD causes RPE apoptosis, leading to cell loss and degeneration.

### 3.4. Prom1 Gene Expression in Human and Mouse Single-Cell RNA-Seq Datasets

To confirm *Prom1* expression in *human* and *mouse* RPE single-cell datasets, we used Spectacle, an interactive web-based resource for exploring published single-cell RNA-sequencing data [28]. Spectacle can identify which retinal cell type expresses a gene of interest, detect transcriptomic subpopulations within a cell type, and perform differential expression analyses. We used a *human* macular and peripheral RPE single-cell RNA- and ATAC-sequencing dataset to analyze *Prom1* gene expression in *human* RPE by region and subpopulation [38]. The Spectacle interactive system was used to visualize graph-based clusters showing subpopulations of *human* RPE cells and their cluster identity in a dimensionality reduction plot where the dimensions are based upon the pre-computed values using uniform manifold approximation and projection (UMAP) [38]. Clusters showing single RPE cells (RPE_0–RPE_4) are shaded in different colors in the dimensionality reduction plot [38]. To determine if the *Prom1* gene is expressed in these various human RPE subpopulations, Spectacle was used to visualize gene expression as violin plots. Our analysis depicts the expression of the *Prom1* gene present in each *human* RPE cluster (Figure 9A). Since *Prom1* expression levels varied between these single-cell RPE clusters, it confirms heterogeneity within RPE tissue from *human* donors. We examined if the *Prom1* gene was enriched in *human* RPE cells across macular and peripheral regions of the retina and found comparable levels of *Prom1* expression in both areas (Figure 9B). These observations demonstrate that across anatomic regions, subpopulations of RPE have similar levels of the *Prom1* gene. To compare *Prom1* expression in *mouse* RPE, we explored other *mouse* RPE datasets (GSE263427 and GSE236220) and found *Prom1* expression in them, confirming the presence of the *Prom1* gene in *mouse* RPE.

To confirm *Prom1* gene expression in other published *human* RPE datasets, we used a single-cell RNA-sequencing dataset of *human* iPSC-derived RPE cells generated from patients with GA and age-matched healthy individuals [39]. In this study, the authors used iPSC-derived RPE cells for disease modeling because this approach has been widely used to demonstrate disease phenotypes, especially those observed in macular dystrophies and AMD [40,41]. The iPSC-RPE cells were characterized based on the expression levels of genes with known associations with GA, including C3, CFH, CFHR1, APOE, and HTRA1, to confirm that GA patient-derived RPE cells are accurate disease models in vitro without aging these cells in culture [39]. Interrogating this dataset, we found robust *Prom1* expression in all subpopulations of *human* RPE cells (Figure 9C). However, this plot does not directly indicate which clusters correspond to AMD or controls. The dimensionality reduction plot using the same dataset was used to distinguish AMD RPE from control RPE clusters (. Mapping the AMD RPE cells to their respective control conditions using Spectacle showed varied *Prom1* expression in a few control cells vs. AMD RPE (i.e., RPE2, RPE3, and RPE4), highlighting *Prom1*’s important role in maintaining RPE function and health, and suggesting that its dysregulation can have significant implications for RPE homeostasis, particularly in the context of AMD (Figure 9D). To assess if the changes in *Prom1* expression in control vs. AMD correlate with other well-known AMD-risk genes, we interrogated the dataset for CFH gene expression, a major AMD-related risk locus [42]. CFH gene expression varied in a few control cells compared to AMD RPE cells (RPE2, RPE4, RPE5), confirming CFH’s involvement in RPE function.

The retina’s resident immune cells, including microglia, support RPE cell function by phagocytosing POSs and lipids, thereby protecting the RPE during outer retinal degenerative conditions [43]. Recently, a unique profile of microglia was uncovered at RPE atrophic sites in *mouse* and *human* models of retinal degeneration [44]. To examine whether *Prom1* is expressed in retinal microglia, we interrogated published databases in Spectacle and found that the *Prom1* gene is highly expressed in activated and proliferating activated microglia (heatmap, Figure 10A), which are known to phagocytose shed POSs at the site of damage and maintain retinal health [45]. Both light and dark conditions influence microglial behavior in the retina differently [45]. Darkness-induced microglial behavior has neuroprotective effects, especially during retinal degeneration. Our analysis showed that activated microglia in darkness in the degenerating retina have high levels of *Prom1* (Figure 10B), suggesting that *Prom1* can mediate microglial behavior in light and dark conditions. Interrogation of a separate *mouse* database [46] confirmed *Prom1* gene expression in microglial cells from normal *mouse* retinas with high expression of initial undamaged microglial markers (MG0) and other clusters of microglia (i.e., lcMG1-3) from light-damaged retinas (Figure 10C). Other small clusters of microglia (srMG) also showed low *Prom1* expression, suggesting that phenotypically heterogenous microglia with unique functional specializations at different locations in the *mouse* retina express *Prom1*.

## 4. Discussion

Despite emerging robust data demonstrating that *Prom1* regulates RPE homeostasis [17,18], direct evidence of its presence in *mouse* RPE in situ has been lacking until now. This study convincingly shows that *mouse* RPE expresses the *Prom1* gene in situ, at least to a sufficient level, to impact key RPE processes, including waste removal by autophagy and stabilizing lysosomal activity. While *Prom1* mRNA expression is significantly higher in the *mouse* PR inner segments, the substantially greater number of PRs than RPE cells in the *mouse* retina [47] contributes partly to this observation. *Prom1*’s function in RPE is distinct from its role in PRs. In PRs, *Prom1* is primarily involved in the structural organization of the OS and is localized to the disk membrane [6,48]. This localization supports its role in maintaining PR integrity and function. In contrast, in RPE cells, *Prom1* is not a membrane-bound protein; its mRNA localizes to the perinuclear region, and we have shown that *Prom1* functions as a signaling molecule within the cytoplasm to regulate autophagy and other intracellular processes integral to *mouse* RPE cell survival. These observations are consistent with our previous findings of cytoplasmic localization of *Prom1* in *human* RPE cells [17]. *Prom1* has been observed translocating from the membrane to the cytoplasm in response to high glucose levels, indicating that the cellular environment and metabolic state significantly influence *Prom1*’s localization [3]. *Prom1’s* trafficking to lysosomes occurs through its physical interactions with cytosolic histone deacetylase 6 (HDAC6) [49]. Our previous studies in *human* RPE cells revealed that cytosolic *Prom1* interacts with autophagy-related proteins such as p62 and HDAC6, playing a crucial role in autophagosome biogenesis, followed by its trafficking to lysosomes [17]. Furthermore, our recent findings show that the loss of *Prom1* in *mouse* RPE cultures disrupts autophagy and induces epithelial–mesenchymal transition, underscoring *Prom1*’s pivotal role as a regulator of cell-autonomous RPE homeostasis [18]. Collectively, these studies establish that *Prom1* predominantly localizes in the cytoplasm under resting conditions in RPE cells and is essential for RPE waste removal, a function markedly distinct from its well-known role at the cell surface.

This study’s novel finding is the presence of *Prom1* in RPE mitochondria. This suggests a role in mitochondrial dynamics, energy metabolism, and apoptosis. This finding could have implications for understanding metabolic disorders and degenerative diseases affecting the RPE. The localization studies expand our knowledge of *Prom1*’s functions and interactions within RPE cells, shedding light on RPE cellular processes and diseases.

Although *Prom1* mutations are linked to macular diseases caused by RPE dysfunction, the precise role of *Prom1* in RPE homeostasis remains unclear. We show that AAV2/1-Cas9-*Prom1*-gRNA-mediated targeting of *Prom1* in *mouse* RPE in vivo leads to RPE abnormalities and secondary PR degeneration. Retinal section analysis revealed patches of caspase-3 positive RPE cells, confirming RPE degeneration and apoptosis due to loss of *Prom1* function, mirroring the RPE degeneration observed in patients with GA. These findings provide insights into the pathophysiology of RPE degeneration in macular diseases and suggest that an RPE-specific *Prom1*-KO mouse model will enhance our understanding of *Prom1*’s cell-autonomous role in maintaining RPE health and homeostasis. This mouse model has been recently created in the laboratory and phenotype studies are underway. The global *Prom1*-KO *mice* of the pure C57BL/6J background showed complete PR degeneration by P20. However, the retinal degeneration in the C57BL/6xCBA/NSlc *mice* was significantly slower, demonstrating that retinal degeneration in *mouse* models depends on genetic background [50]. Although global KO *mouse* models are useful for studying *human* disease, tissue-specific gene targeting is required to understand the specific contribution of *Prom1* in various retinal cell types and the molecular mechanisms associated with diverse disease phenotypes.

*Prom1*-related retinal diseases, although relatively rare, significantly impact vision. The prevalence of these diseases varies based on specific phenotypes and genetic variants [19]. Most studies show that early macular involvement, PR degeneration, and RPE atrophy are common features of *Prom1*-related inherited retinal disease phenotypes [21]. *Prom1* is essential for maintaining the expression of ABCA4 and RDH12 in *mouse* RPE, which is crucial in regulating the visual cycle and preventing toxic bisretinoid accumulation [50]. We have shown that loss of *Prom1* in *mouse* RPE in vitro impairs autophagy flux, activates mTORC1, and decreases transcription factor EB (TFEB) activity, leading to RPE defects similar to aAMD [18]. Furthermore, the loss of *Prom1* function due to a pathogenic *Prom1* mutation, combined with the heterozygous ABCA4 mutation, exacerbates RPE dysfunction, resulting in granular mottling and macula alterations, providing evidence for their shared pathophysiology [20]. Loss of *Prom1* function may lead to lipofuscin accumulation and other toxic RPE metabolites through molecular crosstalk involving ABCA4 and mTORC1. The question remains: how does *Prom1* participate in the expression of these genes? Since *Prom1* is not a transmembrane protein in RPE, it cannot transduce the extracellular information intracellularly, but as a cytoplasmic protein, we have shown it can regulate numerous RPE cellular processes involving degradative pathways. Therefore, additional studies are needed to explore how their interplay impacts downstream target genes and upstream regulatory pathways, affecting RPE cell fate and disease outcomes.

It remains unclear whether *Prom1*-associated retinal diseases originate primarily in the RPE or the PRs with secondary RPE damage. A recent study showed that *Prom1*-null mutations in *Xenopus laevis* cause RPE dysfunction, which precedes PR degeneration [51]. This study uncovered evidence of subretinal drusenoid-like deposits, thinning of RPE, and RPE atrophy in *Prom1*-null *frogs*, suggesting that the loss of *Prom1* primarily impairs RPE function. These observations challenge the previous conclusions that disrupted outer-segment morphogenesis was the primary cause of retinal degeneration associated with *Prom1* loss in a *mouse* model [51,52]. Patients with the *Prom1* R373C mutation (autosomal dominant Stargardt-like) exhibit macular dystrophy with three distinct phenotypes—central GA, multifocal GA, and bull’s eye maculopathy—suggesting an RPE cell-autonomous function of *Prom1* in the *human* retina [24]. A recently identified loss-of-function *Prom1* variant (c.1354dupT) has shown a multi-phenotypic effect linked to cone–rod dystrophy and retinitis pigmentosa (RP) [53]. This cohort suggests that *Prom1*-related pathology may also confer an RP phenotype. Interestingly, a cohort of young patients with this newly identified *Prom1* variant displayed alterations in the RPE with a preserved PR layer. Older patients with more advanced conditions exhibited both PR and RPE degeneration, suggesting that in these patients, the disease initially affects the RPE [53]. Additionally, this study reported the early onset of an autosomal recessive form of STGD4, which differs from the late-onset autosomal dominant manifestation of STGD4 dystrophy. This confirms extensive genetic and phenotypic heterogeneity within *Prom1*-related inherited retinal diseases [19,54]. Some mutations lead to a complete loss of function, while others may result in a partially functional protein, leading to different degrees of retinal and RPE dystrophies [55,56]. Modifier genes, epigenetic modifications, and other genetic factors can influence the heterogeneity of disease presentation, making diagnosis and management challenging.

We examined published *human* RPE single-cell RNA-sequencing datasets using Spectacle and found the expression of the *Prom1* gene in *human* RPE. *Prom1* is expressed in the *human* retina’s macular and peripheral regions, indicating its fundamental role in maintaining RPE function. Importantly, we observed variations in *Prom1* expression between distinct clusters of control RPE versus AMD RPE. These variations need further exploration to understand how *Prom1* function influences RPE dysfunction and degeneration in aAMD and its potential significance in disease pathogenesis and RPE homeostasis. Analysis of a *mouse* retinal single-cell RNA-sequencing dataset revealed the presence of *Prom1* in RPE, amacrine cells, bipolar cells, early and late retinal progenitor cells, and Müller glia, suggesting *Prom1*’s potential involvement in regulating retinal development, regeneration, and disease processes [57]. The presence of *Prom1* in the synaptically active amacrine cells suggests its role in integrating and shaping the visual message presented to retinal ganglion cells. *Prom1* in amacrine cells may contribute to retinal circuitry by enhancing the communication between PRs, bipolar cells, and ganglion cells, ensuring efficient visual processing and adaptation to changing environmental conditions. Additional studies are warranted to test *Prom1*’s emerging function in the amacrine cells, retinal progenitor cells, and Müller glia.

We also found *Prom1* expression in advanced and proliferating retinal microglia, specialized immune cells critical for surveillance, phagocytosis, and retinal health. Interestingly, activated microglia in the degenerating retina exhibit high levels of the *Prom1* gene, particularly during darkness, indicating that *Prom1* may be involved in microglial responses to retinal stress. Furthermore, other small microglial populations also express *Prom1*, suggesting its unique functional roles in regulating phenotypically heterogeneous microglia at different retinal locations. The intricate relationship between *Prom1* and microglia underscores the complexity and importance of understanding *Prom1*-related retinal pathologies.

In summary, our study provides comprehensive insights into the multifaceted role of *Prom1* in RPE homeostasis and retinal health. The detailed characterization of *Prom1*’s localization, function, and implications in RPE/retinal health underscores its significance as a therapeutic target. Future research on elucidating the molecular mechanisms of *Prom1*’s interactions with other cellular pathways will be essential for developing effective therapies for aAMD and other *Prom1*-associated retinal diseases. Our findings highlight the importance of *Prom1* as a central mediator in maintaining RPE health and offer promising directions for therapeutic advancements in retinal diseases.

## 5. Limitations

Our study has certain limitations associated with using subretinal injections of AAV2/1.saCas9.*Prom1*-gRNA to target *Prom1* in the mouse RPE in situ. Subretinal injections typically result in the localized delivery of the vector, meaning the AAV2/1.saCas9.gRNA complex predominantly transduces RPE cells near the injection site. The restricted diffusion of the vector from the injection site can lead to uneven distribution, limiting *Prom1* gene knockdown to specific areas. Additionally, the retinal structure and extracellular matrix act as barriers that hinder the widespread dissemination of the vector, resulting in more localized effects. Variability in the subretinal injection procedure, such as differences in injection volume and pressure, can also affect the vector’s spread and the pattern of *Prom1* gene knockdown. Notably, the patchy nature of *Prom1* knockdown and resulting RPE degeneration more closely mirrors the localized areas of RPE loss and dysfunction seen in human GA, a late-stage manifestation of dry AMD. This finding highlights the potential relevance of our model for studying GA progression and therapeutic interventions.

In our study, the eyes injected with *Prom1*-gRNA exhibited cloudy, patchy, and diffuse yellowish lesions by fundus imaging, consistent with RPE cell pathology and/or dysfunction; this was the only direct evidence of lesions observed. However, H&E staining did not reveal classic atrophic lesions. Nonetheless, the observed RPE degeneration, marked by active caspase-3 labeling, indicates significant RPE cell apoptosis, an early indicator of atrophy. Localized RPE dysfunction may also affect electroretinography (ERG) readings, suggesting photoreceptor (PR) degeneration. Further studies, including quantifying PR layer thickness, are necessary to confirm secondary PR loss. While these additional studies are crucial for fully characterizing the phenotypic changes following *Prom1* knockdown in *mouse* RPE in situ, they were not performed in this study.

## Figures and Tables

**Figure 1 cells-13-01761-f001:**
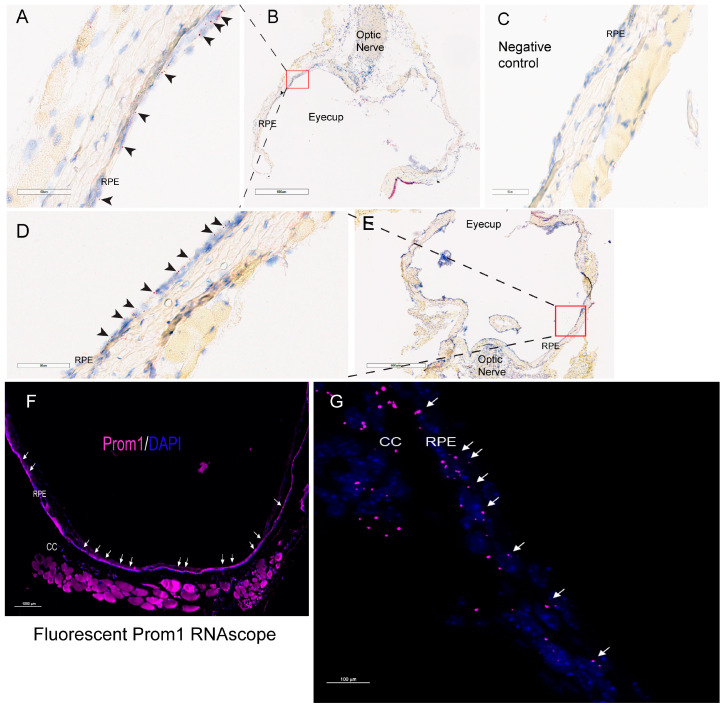
*Prom1* mRNA expression in albino mouse RPE in situ. Representative brightfield images showing chromogenic *Prom1* RNAscope (red) in albino mouse eyecup sections. The sections were counterstained with hematoxylin (blue). (**A**,**B**) The black arrowheads show *Prom1* mRNA expression in *mouse* RPE in situ. The scale bar is 60 μm. (**B**) The brightfield low-resolution image of the eyecup and the red box showing the area of the eyecup used for imaging. The scale bar is 600 mm. (**C**) No *Prom1* labeling in the negative control for *Prom1* RNAscope. The scale bar is 60 mm. (**D**) Imaging from another *mouse* eyecup shows *Prom1* mRNA expression in *mouse* RPE in situ (black arrowheads). The scale bar is 60 μm. (**E**) The brightfield low-resolution image of the eyecup and the red box showing the area of the eyecup used for imaging. The scale bar is 600 μm. (**F**) Representative fluorescent image showing *Prom1* expression in an albino *mouse* eyecup (white arrows). The section was counterstained with DAPI (blue). The scale bar is 1000 μm. (**G**) Confocal high-resolution micrograph of *Prom1* expression in *mouse* RPE (white arrows; purple fluorescence). CC = choriocapillaris. The scale bar is 100 μm.

**Figure 2 cells-13-01761-f002:**
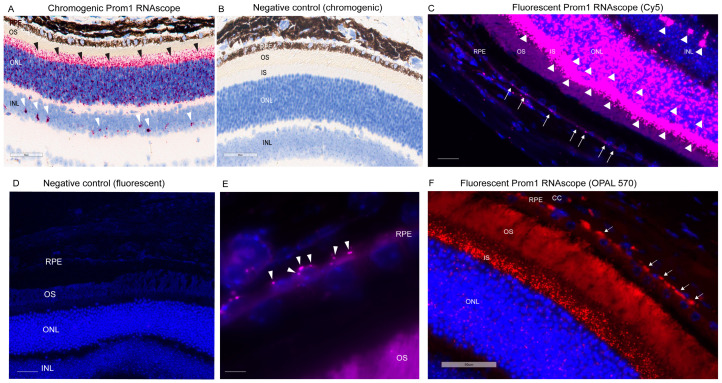
*Prom1* RNAscope in pigmented C57BL/6J *mouse* retinal sections. Brightfield 40X slide scanning micrographs of (**A**) pigmented C57BL/6J retinal sections labeled by chromogenic RNAscope for *Prom1* (red puncta). Labeling is present in the photoreceptor inner segments (ISs) and outer nuclear layer (ONL) (black arrowheads) and a cohort of cells in the inner retina (white arrowheads). The scale bar is 60 μm. (**B**) No labeling was detected in the negative control of chromogenic *Prom1* RNAscope. The scale bar is 60 μm. (**C**) Fluorescent widefield confocal micrographs of *Prom1* labeling with Cy5 fluorophore (magenta puncta) and DAPI counterstain in C57BL/6J retinal sections showing *Prom1* mRNA expression (magenta puncta) in the mouse RPE (white arrows), photoreceptor inner segments (ISs), ONL, and a cohort of cells in the inner retina (white arrowheads). The scale bar is 200 μm. (**D**) Negative control of fluorescent *Prom1* RNAscope in retinal sections shows no *Prom1* labeling. The scale bar is 200 μm. (**E**) Representative high-resolution widefield confocal microscopy using 100X objective showing *Prom1* mRNA expression (magenta puncta) in single RPE cells in situ by a fluorescent RNAscope assay (white arrowheads). Sections were counterstained with DAPI (blue). The scale bar is 200 μm. (**F**) Representative fluorescent 40× Leica slide scanning images showing the presence of *Prom1* mRNA in RPE (white arrows), photoreceptor ISs, and ONL in *mouse* retinal sections. DAPI was used as a counterstain. The scale bar is 50 μm. CC = choriocapillaris.

**Figure 3 cells-13-01761-f003:**
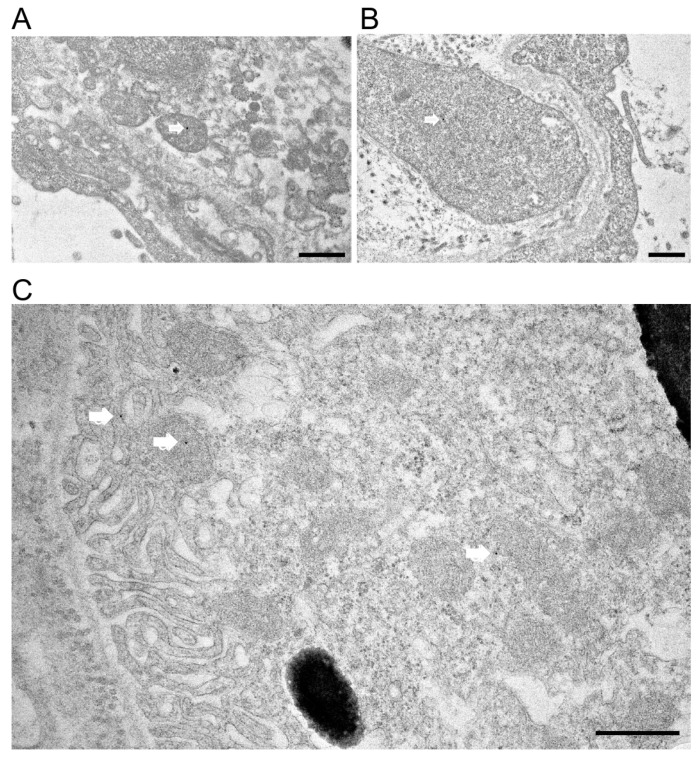
Immunogold transmission electron microscopy in C57BL/6J mouse eyecups. Immunogold TEM of *Prom1* in C57BL/6J *mouse* eyecups. (**A**,**B**) TEM micrographs showing positive *Prom1* labeling in RPE mitochondria (white arrows). The scale bar is 500 nm. (**C**) Immunogold TEM of mouse RPE in situ showing cytoplasmic localization of *Prom1* (white arrow, top left), mitochondria (white arrow, middle left), and in proximity to mitochondria (bottom right). The scale bar is 500 nm.

**Figure 4 cells-13-01761-f004:**
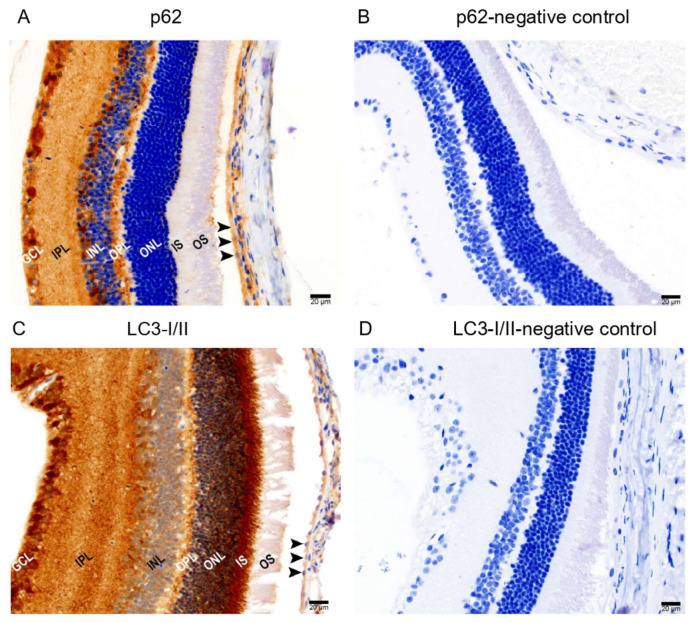
Immunohistochemical detection of autophagy markers in *mouse* retinal sections. Representative brightfield micrographs showing (**A**) positive p62 immunolabeling in albino *mouse* retinal sections and (**B**) a negative control of p62 immunostaining. (**C**) LC3-I/II protein is expressed in *mouse* RPE and other cell types in the retina. (**D**) Hematoxylin was used to stain the nuclei for both p62 and LC3-I/II immunolabeling. The scale bar is 20 μm.

**Figure 5 cells-13-01761-f005:**
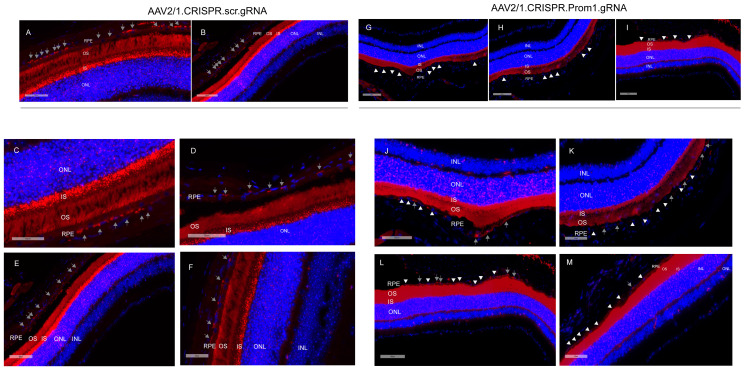
Subretinal injections of AAV2/1.*Prom1*.gRNA leads to *Prom1*-KD in mouse RPE in situ. Representative micrographs of retinal sections of C57BL/6J *mouse* eyes labeled by fluorescent RNAscope for *Prom1* (red puncta). (**A**,**B**) Low-magnification fluorescent micrographs with wider views of retinal sections injected with control AAV2/1.CRISPR.scr.gRNA. Scale bars of 60 μm and 100 μm. (**C**–**F**) Images of various magnifications showing *Prom1* labeling (red puncta) in *mouse* RPE in control sections (gray arrows). Scale bars ranging from 50 μm to 100 μm. The sections were counterstained with DAPI. (**G**–**I**) Fluorescent micrographs with wider views of retinal sections injected with experimental AAV2/1.CRISPR.*Prom1*.gRNA. Scale bars ranging from 80 μm to 90 μm. The white arrowheads show patchy areas where the Prom1 gene has been knocked down in the RPE in situ. (**J**–**M**) Low- and high-magnification micrographs of retinal sections injected with AAV2/1.CRISPR.*Prom1*.gRNA shows patchy *Prom1*-KD (white arrowheads) in RPE, with areas showing unaltered *Prom1* expression (gray arrows). Scale bars ranging from 50 μm to 100 μm.

**Figure 6 cells-13-01761-f006:**
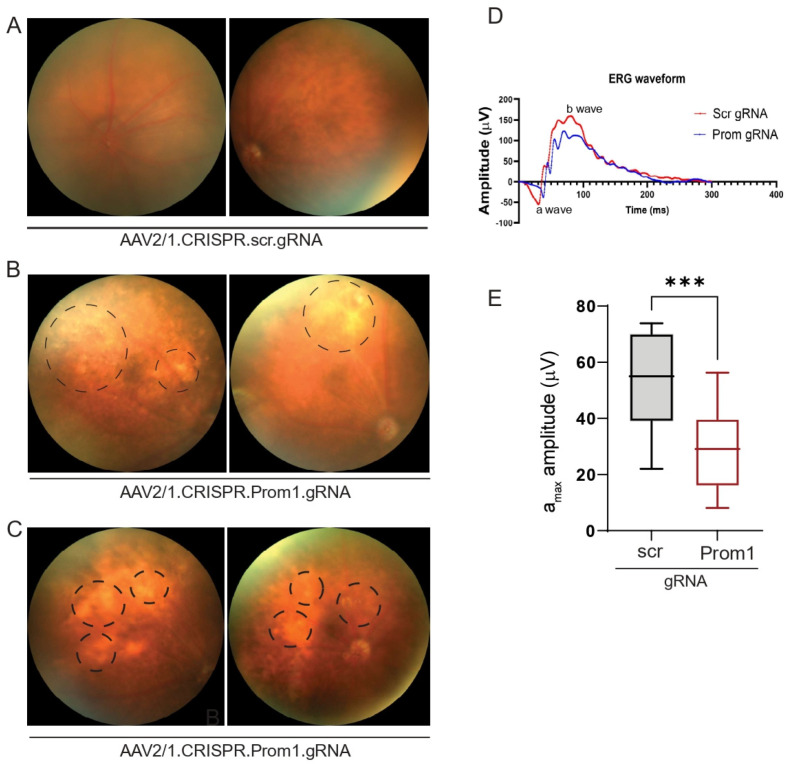
Fundus imaging and ERG of *mouse* eyes with subretinal injection of control or experimental viral vectors. Representative fundus images of *mouse* eyes injected with (**A**) control AAV2/1.CRISPR.scr.gRNA or (**B**,**C**) experimental AAV2/1.CRISPR.*Prom1*.gRNA. Images were obtained after 11 weeks of injection. Circles with dashed lines show areas of RPE degeneration in *mouse* eyes injected with *Prom1*-gRNA. (**D**) Representative ERG waveforms (a- and b-waves) in *mouse* eyes injected with scrambled (scr) or *Prom1* gRNA. (**E**) Quantifying scotopic a-wave ERG responses from *mouse* eyes injected with scr- or *Prom1*-gRNA. ***, *p* value 0.0004.

**Figure 7 cells-13-01761-f007:**
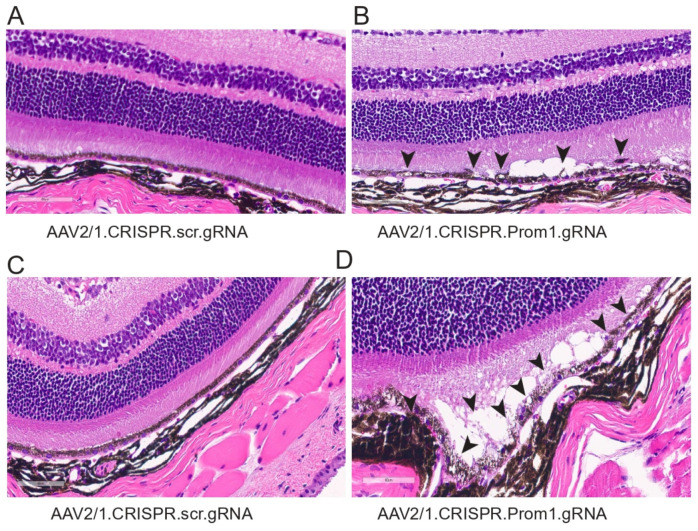
Histology of *mouse* eyes injected with control or experimental viral vectors. Brightfield micrographs of H&E-stained sections of C57BL/6J eyes injected with either (**A**) control AAV2/1.CRISPR.scr.gRNA or (**B**–**D**) experimental AAV2/1.CRISPR.*Prom1*.gRNA. *Prom1* knockdown causes patchy RPE vacuolization and abnormalities with fluid accumulation between the RPE and PRs (back arrowheads)—scale bars for all micrographs, 60 μm.

**Figure 8 cells-13-01761-f008:**
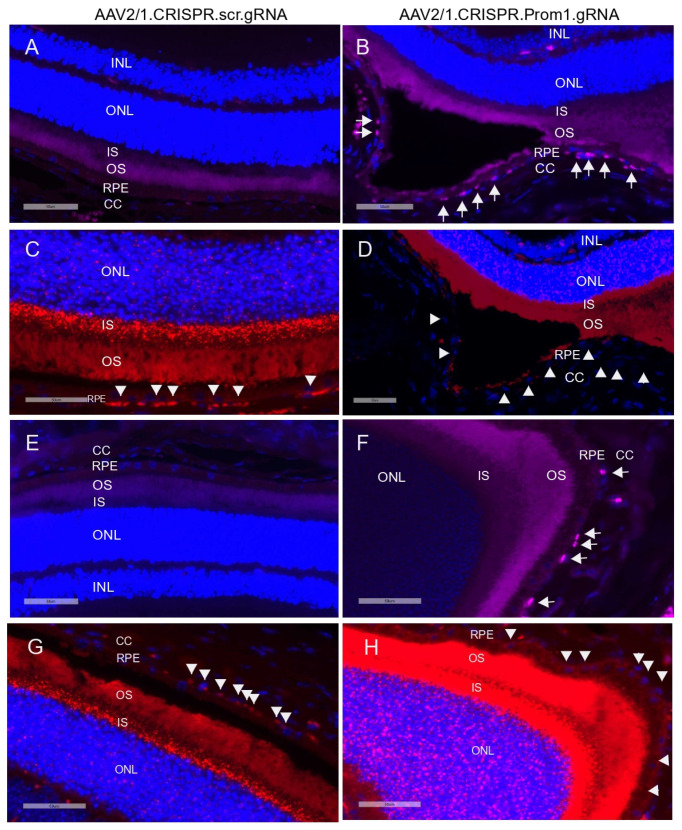
*Prom1* knockdown in vivo causes RPE apoptosis and degeneration. Retinal sections of C57BL/6J mouse eyes injected with (**A**,**E**) control AAV2/1.CRISPR.scr.gRNA or (**B**,**F**) experimental AAV2/1.CRISPR.*Prom1*.gRNA were used to detect active cleaved caspase-3 by immunohistochemistry. White arrows show positive immunolabeling of RPE cells for active caspase-3 in retinal sections obtained from *Prom1*-gRNA-injected eyes but not in RPE from control eyes. Serial sections from the same *mouse* eyes were labeled by fluorescent RNAscope for *Prom1*. (**C**,**G**) *Prom1* labeling (red puncta) was observed in eyes injected with control AAV2/1.CRISPR.scr.gRNA, but its expression was reduced in eyes injected with (**D**,**H**) experimental AAV2/1.CRISPR.*Prom1*.gRNA (white arrowheads). Scale bars for all micrographs, 50 μm.

**Figure 9 cells-13-01761-f009:**
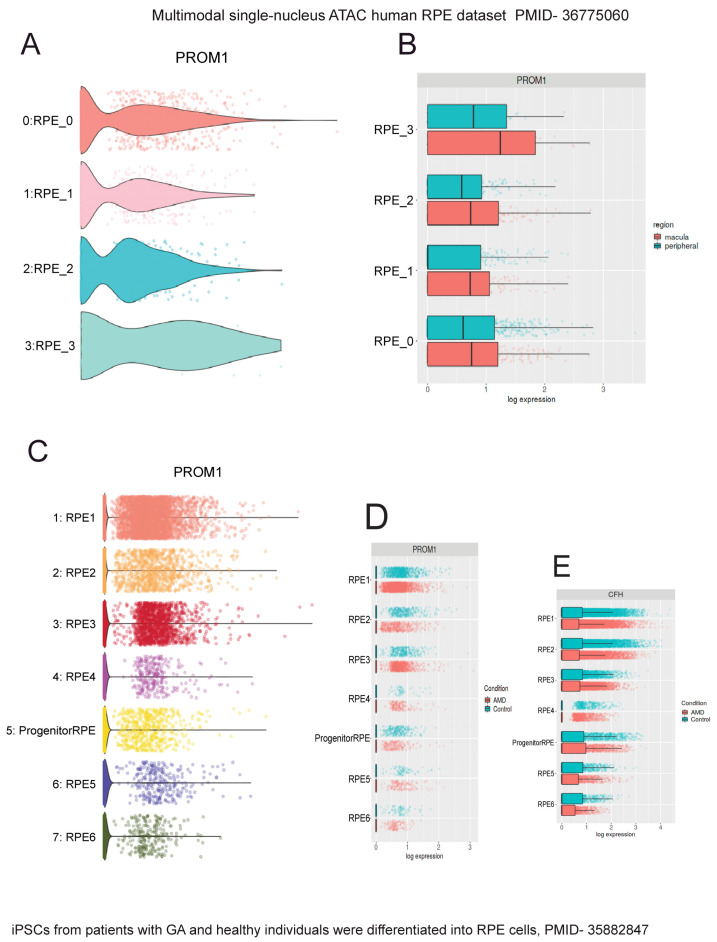
Interrogation of single-cell *human* RPE datasets using Spectacle shows *Prom1* gene expression in *human* RPE. The single-nucleus ATAC *human* RPE dataset (PMID: 36775060) [38] was used to analyze *Prom1* gene expression. (**A**) Violin plot showing *Prom1* gene expression in RPE single-cell clusters (RPE_0 to RPE_3). (**B**) Violin plot showing similar *Prom1* gene expression levels across macular and peripheral regions. The patient iPSC-derived RPE dataset (PMID: 35882847) [39] was used to analyze *Prom1* gene expression. (**C**) Volcano plot showing *Prom1* gene expression in multiple single-cell RPE clusters and progenitor/RPE cells. (**D**) Volcano plot showing *Prom1* gene expression in control vs. AMD RPE. The differences between red (AMD) and blue (control) bars in some RPE subsets (RPE2, RPE3, and RPE4) suggest changes in *Prom1* gene expression between control and disease conditions. (**E**) Plot showing AMD-risk HTRA1 gene expression levels in control vs. AMD conditions.

**Figure 10 cells-13-01761-f010:**
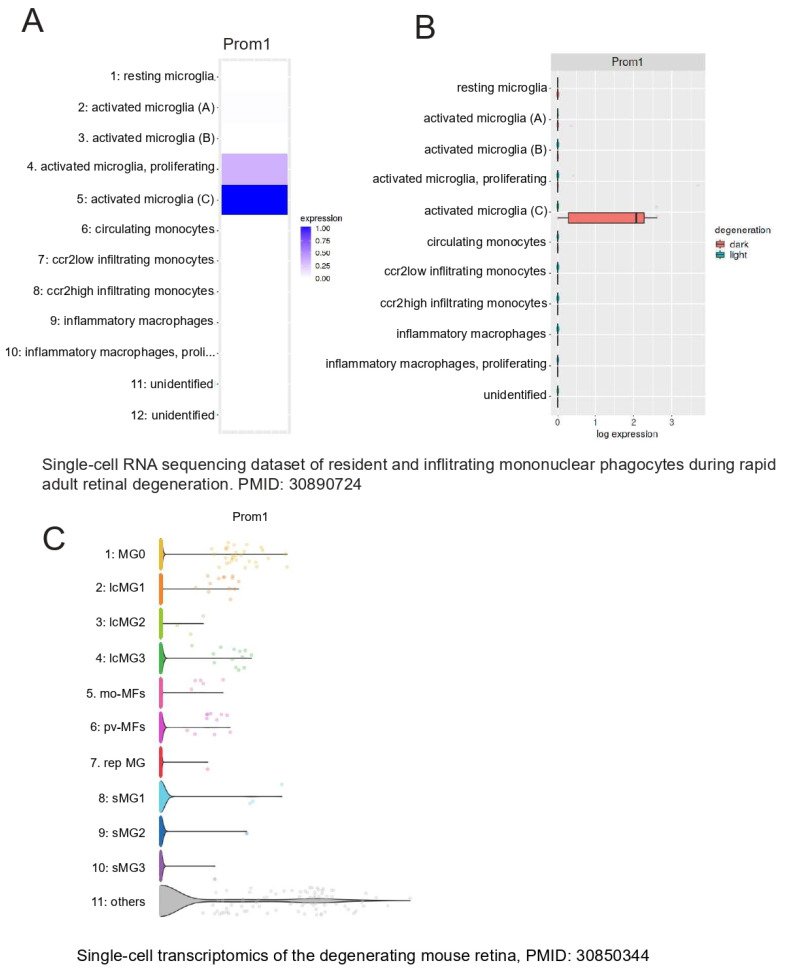
*Prom1* gene expression in *mouse* retinal microglia. Analysis of a *mouse* single-cell RNA dataset (PMID: 30890724) [45] using Spectacle. (**A**) Heatmap showing the *Prom1* gene is highly expressed in proliferating and activated microglia. (**B**) Plot showing *Prom1* gene expression in a subset of activated microglia in the degenerating *mouse* retina under dark conditions. A mouse retinal degeneration (induced by light damage) dataset (PMID: 30850344) [46] was used to analyze *Prom1* gene expression in adult retinal microglia. (**C**) The volcano plot shows the levels of *Prom1* gene expression in undamaged microglia (MG0), other clusters of microglia (lcMG), and small clusters of microglia (sMG).

## Data Availability

The data supporting this study’s findings are available from the corresponding author upon reasonable request.

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
