# Peer review of "Prominin-1 Knockdown Causes RPE Degeneration in a Mouse Modelâ€"

_cells, 2024, doi:10.3390/cells13211761_

Round 1
Reviewer 1 Report
Comments and Suggestions for Authors
The manuscript “Prominin-1 knockdown causes RPE degeneration in a mouse model” by Sujoy Bhattacharya et al. examines primarily the expression of the gene encoding the protein Prom1. Mutations in Prom1 are associated with retinal degenerations in humans, making the expression and function of the protein of interest in the community. A curious point here is that Prom1 is a transmembrane protein located to the cytoplasm, largely shown in the groups previous report. The localization is also cell-type dependent, being in the photoreceptor outer segments but also the cytoplasm of the RPE. This unique dual localization, membrane in some cells, but cytoplasmic in others requires careful trafficking studies, which are not presented here. The actual function of the protein remains unknown and is not addressed in this study. Overall, as detailed in my specific comments below, I found the presentation below expectations and the data descriptive. Given the complexities of the protein, protein data needs to be included with the gene expression data.
Specific Comments:
1. Figure 1. The magnification of the images are not the same, which limits audience understanding and interpretation. Further, the panels are not presented in the same orientation (A, B then C). The size markers should be the same for each image in the figure, and needs to be added to D and E.
2. Figure 2 same problems (See figure 1 comments).
3. Figure 3. 1 gold particle for each image is not convincing. This does not fit with the expression levels in Fig 4. The authors want us to believe Prom1 functions in the mitochondria, but most of the mitochondria in the image are not labeled. Coupled with the just one gold particle each in 2 different mitochondria leads me to think this is an artifact at best. Is there another example of a mitochondrial protein with similar minimal expression levels?
4. Fig 4. See comments for Fig 1, again different magnifications. A perhaps also different exposures as well and panel C does not look like it belongs with the others for whatever reason.
5. Fig 6, see comments for Fig 1. Different magnifications and no consistent magnification bars. This is not interpretable as presented.
6. Figure 7 we finally get size bars, but all the pictures are at different magnifications, and different exposures. The data is not interpretable this way.
7. Figure 8. See all of the comments above (different magnification and exposure), and add that the pictures are now blurry as well. Not presented appropriately for the audience to interpret.
8. Figure 9. Induced pluripotent stem cells (iPSC) from geographic atrophy patients compared to healthy people leads to host of problems not addressed here. To start with, iPSC are fetal in nature, by definition, so unless the authors ‘aged’ them somehow they are not a model of an aging disease. Secondly, GA is not genetic, so why do the authors think there will be a difference between normal and GA RPE from iPSC’s maintained under the same conditions? How does this fit GA, please explain this hypothesis. Is the purpose of figure 9 as stated ‘to show Prom1 is expressed in human RPE’?
Reviewer 2 Report
Comments and Suggestions for Authors
The authors present an interesting manuscript describing the knock-down of Prom1 in RPE cells in mice. This work has the potential to shed new light on the role of Prom1 in the RPE. Several points to address in this paper:
1. Figure 1 - Panel A. Inset needs to be of better quality/contrast.
2. Figure 1 - Panel B. Eyecup looks mangled. Need a better picture. Where is the location of the optic nerve?
3. Figure 1 - Panel C. Label RPE. Also, is the scale bar the same as in panels A and B?
4. Figure 3. The mitochondrial labeling in the EM is not very convincing. As such, it does not support the authors' claim regarding the mitochondrial localization of Prom1. It would be better to do mitochondrial isolation/enrichment from the RPE and co-stain for mitochondrial markers and Prom1.
5. Figure 4 - this figure does not really add anything to the manuscript. It is already well known that autophagy components are present in the RPE. It appears that the authors' previous work already shows the importance of Prom1 in regulating autophagy.
6. Figure 7 should appear before Figures 5 and 6.
7. Figure 5 - based on figure 7, it would seem that the lesions are in the areas of the sub retinal injection of the vector. Is that correct? What about in eyes with multiple lesions? Were there multiple injections?
8. Figure 5 - what about b-wave of the ERG? Also, how many mice were in each arm? (i.e. need to put sample size for the histogram).
9. Figure 6 - Was lipid staining done to determine if the subretinal clear spaces were fluid or lipids. Given yellow appearance on the fundus photos, this should be done.
10. Figure 8 - at what time point after vector injection was the caspase 3 measured? If one waited longer, would one start to see caspase 3 activation in the retina?
11. A general comment - the authors do show that the knockdown of Prom1 in the RPE is giving an RPE phenotype. Calling this atrophy maybe a bit of a stretch, as they are not showing atrophic lesions on any of their imaging. Also, to say that there is secondary PR degeneration is not supported by their data. The ERG findings could just be the result of localized RPE dysfunction. To truly state that there is secondary PR loss, they need to do better quantification of the PR layer.
Round 2
Reviewer 1 Report
Comments and Suggestions for Authors
The manuscript by Sujoy Bhattacharya has been revised according to the reviewer comments, improving the manuscript significantly. The majority of concerns in the initial review involved data presentation and figures. These issues have been dealt with a variety of success and failure. For example, the labeling in scale bars for Fig 2 C, D, and E as well as in Figure 5 are impossible to read. Best solution is to remove labels in/on scale bars, and state the size in the legend as the authors have done already. For complex figures (2 and 5), just tell the audience which groups have which size bars. For fig 2, the bars should look similar in the various panels.
Reviewer 2 Report
Comments and Suggestions for Authors
My concerns have been addressed. Thank you.
Author Response
We appreciate your time and effort in providing valuable suggestions to improve our work.